# Numerical Study of Flow Downstream a Step with a Cylinder Part 2: Effect of a Cylinder on the Flow over the Step

Milad Abdollahpour [1,*], Paola Gualtieri [1], David F. Vetsch [2] and Carlo Gualtieri [3,*]

1 Department of Civil, Architectural and Environmental Engineering, University of Naples Federico II, 80125 Napoli, Italy
2 Laboratory of Hydraulics, Hydrology and Glaciology, Department of Civil, Environmental and Geomatic Engineering, ETH Zurich, 8093 Zurich, Switzerland
3 Department of Structures for Engineering and Architecture, University of Naples Federico II, 80125 Napoli, Italy
* Correspondence: m.abdolahpour@yahoo.com (M.A.); carlo.gualtieri@unina.it (C.G.)

**Abstract:** In this study, divided into two parts, the effect on a two-dimensional backward-facing step flow (BFSF) of a cylinder placed downstream of the step was numerically investigated. While in Part 1, the numerical simulations carried out without the cylinder were validated using the available literature data, in Part 2 the effect of the cylinder was investigated. In the laminar regime, different Reynolds numbers were considered. In the turbulent regime, the effects on the flow structure of a cylinder placed at different horizontal and vertical locations downstream of the step were comparatively studied. When the cylinder was positioned below the step edge mid-plane, flow over the step was not altered by a cylinder. However, in other locations of a cylinder, the added cylinder modified the structure of flow, increasing the skin friction coefficient in the recirculation zone. Furthermore, the pressure coefficient of the bottom wall increased immediately downstream of the cylinder and farther downstream of the reattachment point and remained stable in the flow recovery process. Moreover, the presence of the step significantly influenced the dynamics of the vortex generation and shedding leading to an asymmetric wake distribution.

**Keywords:** backward-facing step flow (BFSF); flow past a cylinder; computational fluid dynamics; laminar flow; turbulent flow





## 1. Introduction

Backward-facing step flow (BFSF) is one representative separation flow model in fluid mechanical problems, hydraulic engineering, and environmental hydraulics [1]. In Part 1 of this two-part paper [2], the most important characteristics of the backward-facing step flow, basic mechanisms, and experimental and numerical topics were reviewed. In recent years, the control methods of recirculation flow downstream of the backward-facing step emerged [3–7]. Flow past a cylinder as a benchmark in fluid mechanics could be controlled flow over a backward-facing step, and understanding the interactions of the step and the cylinder is important. In environmental applications, cylindrical obstacles such as large pieces of wood may be trapped near the step, altering the turbulent properties of the flow. Recirculation zones and transverse flows downstream of the step typically play an important role in stream ecology as they can increase the residence time of solutes, matter, and sediments to enhance deposition processes.

The controlling parameters of BFS may include various effects on separation, reattachment length, near-wall pressure coefficient, wall skin friction, velocity field, turbulent kinetic energy, and many others [8–11]. Controlling the BFSF with new geometric designs has been studied in recent years, such as a method based on suction or blowing downstream of the step by Uruba et al. [12]. Such geometric modifications reduced the length of the separation zone. More recently, the flow over an inclined step has been also investigated [13–21].

Those studies showed that the size of the recirculation zone increases as the step angle increases. The control of the separation region in the BFSF using rib upstream of the step was already studied [22–24]. Those results demonstrated that a single rib upstream of the step is very effective in changing the average streamwise velocity profiles and turbulent fluctuations, as well as in decreasing the reattachment length.

In recent years, cylinders are also used in the modification of BFS flows [25–27]. The cylinder creates a large drag due to the periodic separation and causes some differences in the pressure between the downstream and upstream. Characteristics of this flow are the separation and reattachment of the boundary layers, wake interactions, vortex breakdown, and merging [28]. The flow pattern around the cylinder changes as the Reynolds number based on cylinder diameter increases [29–32]. Heat transfer and fluid flow characteristics over a backward- or forward-facing step with the insertion of a cylinder have received some attention in the literature [25,33–35]. Kumar et al. [25] studied the effect of a cylinder on separated forced convection at a BFSF. Their study focused on heat transfer enhancement of BFSF laminar flows by using a single adiabatic circular cylinder. Chen et al. [36] designed a cylinder to test the effect of the cylinder on the temperature gradient in the BFSF. They applied the Lattice Boltzmann Method (LBM) in Reynolds range limited to a maximum value of Re = 200. The results have shown that inserting the cylinder enhances heat transfer and leads to a reduction in the intersection angle between the velocity vector and the temperature gradient. Selimefendigil and Oztop [37,38] designed a rotating cylinder in the BFS ferrofluid flow. Their results showed that the averaged heat transfer increases as the Reynolds number increases, and the rotating cylinder enhanced the heat transfer. The studies of a cylinder placed downstream of the step are mostly limited to heat transfer and magnetohydrodynamics. As emerged from the literature review, important gaps exist in the knowledge of the effect of different locations of the cylinder on the flow and turbulent characteristics of BFSF.

The objective of this study is to present the effect of the cylinder on the 2D BFS flow structure in the laminar and turbulent regimes. In Part 1, the numerical simulations carried out without the cylinder were validated using the available literature data. In Part 2, the effect of the cylinder at different horizontal and vertical locations downstream of the step was investigated. Part 2 is organized as follows: Section 2 presents the numerical modeling set-up including geometry mesh generation and boundary conditions for the backward-facing step with a cylinder. Section 3 reports the main results of the numerical study, such as reattachment length, recirculation zone, velocity profile, skin friction and pressure coefficient, turbulent kinetic energy, cylinder wakes, and how the cylinder interact with these characteristics. These findings are discussed in Section 4. Finally, the conclusions are presented in Section 5.

## 2. Numerical Modeling Set-Up

### 2.1. Laminar Flow

A two-dimensional backward-facing step with a cylinder, namely BFSF 2, was considered following the geometry studied in Part 1 [2]. A cylinder with a diameter (D) was set at one cylinder-diameter distance from the step edge (x = D) in the x-direction. The top half of the cylinder was above the top surface (mid-plane) of the step. The sketch of the BFSF with a cylinder in the laminar regime is presented in Figure 1.

As for the BFSF 1, the structured rectangular hexahedral mesh was considered for BFSF 2. Figure 2 shows the computational mesh with decreasing cell size towards the walls and near the cylinder.

As for the BFSF 1, mesh independence was checked for BFSF 2 by comparison of the dimensionless reattachment lengths $Lr_1/h$ and $Lr_2/h$, in a $Re_h = 544$ at five different sizes (see Table 1). It was found that the computed results were independent of the number of cells. The aim was to keep the cell numbers of BFSF 2 close to that of BFSF1 (129,200). The mesh resolution of BFSF 1 was chosen to account for the cylinder and provide consistent results for comparison. A total number of 129,600 cells was selected.

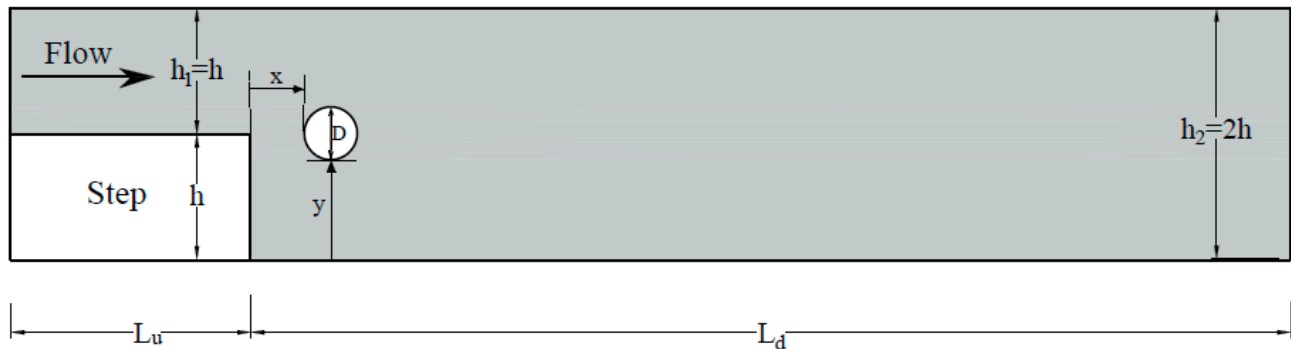

**Figure 1.** Schema of BFSF with a cylinder in laminar flow simulations (BFSF 2) (not to scale).

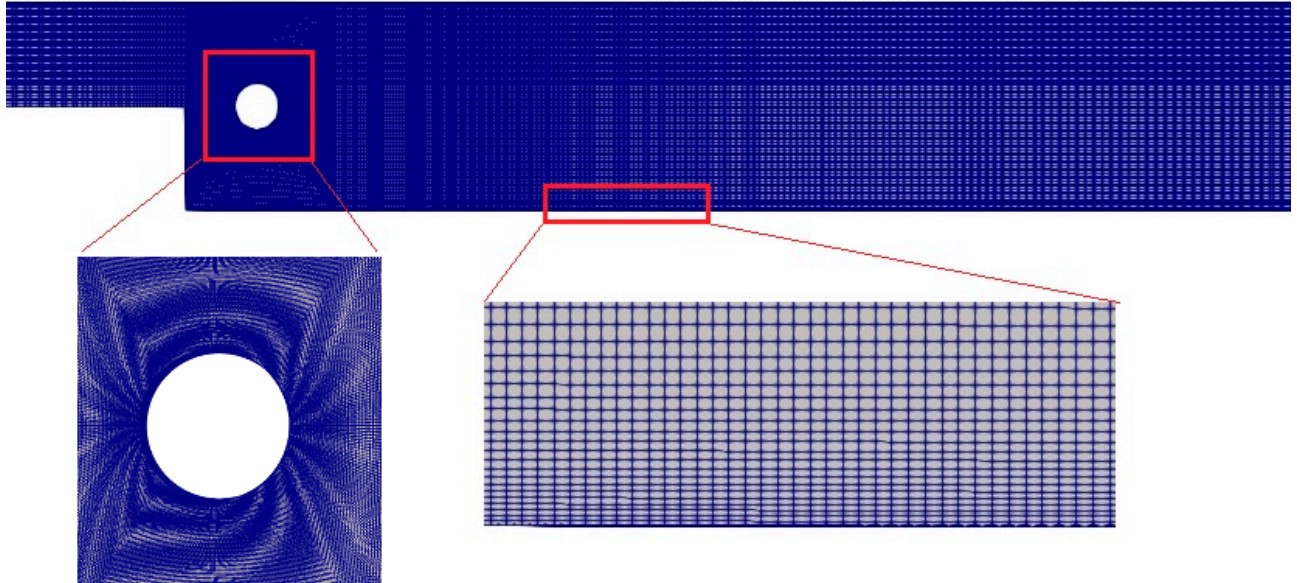

**Figure 2.** The general view of the grid and zoom of the grid in the vicinity of the cylinder and bottom wall.

**Table 1.** Grid independence test results for BFSF 2.

| Grid No. | Number of Cells | $Lr_1/h$ | $Lr_3/h$ |
|----------|-----------------|----------|----------|
| Mesh 1 | 72,000 | 0.815 | 17.15 |
| Mesh 2 | 104,400 | 0.832 | 17.30 |
| Mesh 3 | 129,600 | 0.855 | 17.50 |
| Mesh 4 | 132,600 | 0.860 | 17.55 |
| Mesh 5 | 148,500 | 0.865 | 17.60 |

The boundary conditions assigned at the inlet, the outlet, and the walls were those for BFSF 1. It is noted that in the BFSF 2, the cylinder was assumed to be a wall. The Reynolds number based on the step height (h) was defined as $Re_h = \frac{Uh}{\upsilon}$ and based on cylinder diameter (D) was defined $Re_D = \frac{UD}{\upsilon}$, where U is the inlet flow velocity. Table 2 lists the values of Reynolds numbers in laminar flow.

### 2.2. Turbulent Flow

In the turbulent regime, a cylinder with different horizontal and vertical locations downstream of the step were considered. Two series of numerical tests were studied. In series (I), a cylinder with a diameter (D) was set at a different distance from the step edge, in

the x-direction. In series (II), a cylinder was set at one cylinder-diameter distance from the step edge at different locations in the y-direction. The Reynolds number based on the step height (h) was $Re_h = 9000$ and based on cylinder diameter (D) was $Re_D = 2015$. Sketches of the BFSF and its configurations in the turbulent regime are presented in Figure 3 and Table 3.

**Table 2.** Reynolds numbers based on step height ($Re_h$) and cylinder diameter ($Re_D$) in laminar flow in the backward-facing step flow with a cylinder.

| Run | $Re_h$ (Step Height) | $Re_D$ (Cylinder Diameter) | x | y |
|---|---|---|---|---|
| BFSF 2—L1 | 75 | 15 | | |
| BFSF 2—L2 | 158 | 28 | | |
| BFSF 2—L3 | 336 | 55 | | |
| BFSF 2—L4 | 420 | 70 | 1 D | $h - \frac{D}{2}$ |
| BFSF 2—L5 | 544 | 87 | | |
| BFSF 2—L6 | 672 | 108 | | |
| BFSF 2—L7 | 755 | 120 | | |

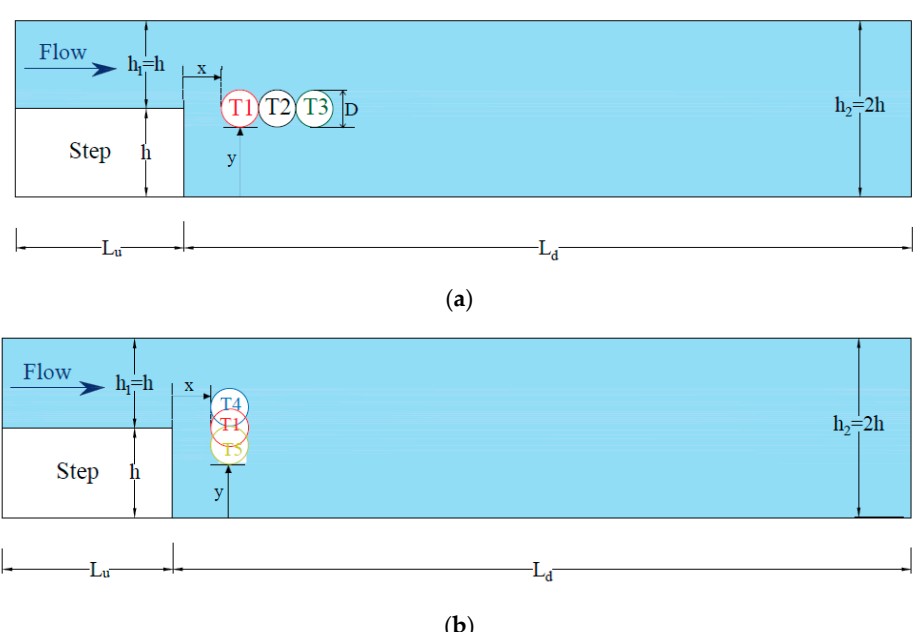

(**a**)

(**b**)

**Figure 3.** Sketches of the BFSF with a cylinder in turbulent flow simulations (not to scale). (**a**) Cylinder was set at a different distance from the step edge in the x-direction (**b**) Cylinder was set at one cylinder-diameter distance from the step edge at different locations in the y-direction.

**Table 3.** BFSF 2 configurations of the turbulent regime.

| Run | $Re_h$ (Step Height) | $Re_D$ (Cylinder Diameter) | x | y |
|---|---|---|---|---|
| BFSF 2—T1 | 9000 | 2015 | 1 D | $h - \frac{D}{2}$ |
| BFSF 2—T2 | 9000 | 2015 | 2 D | $h - \frac{D}{2}$ |
| BFSF 2—T3 | 9000 | 2015 | 3 D | $h - \frac{D}{2}$ |
| BFSF 2—T4 | 9000 | 2015 | 1 D | $h$ |
| BFSF 2—T5 | 9000 | 2015 | 1 D | $h - D$ |

As the classical backward-facing step (BFSF 1), the structured rectangular hexahedral mesh was considered for the BFSF 2 with a total number of 129,600 cells. The boundary conditions were considered the same as BFSF 1. A wall boundary condition was assigned for the cylinder. The boundary conditions were considered the same as BFSF 1.

The BFSF model was validated in Part 1 of the paper [2], and it was found that the standard k-ε model performed best for the current cases where the focus is on the flow recirculation behind the step. Thus, the same turbulence model was used for the study of the effect of a cylinder placed downstream of the step.

## 3. Results

### *3.1. Laminar Flow*

#### 3.1.1. Recirculation Zone and Reattachment Length

The most important characteristics of flow over the step are flow separation and reattachment [39]. The adverse pressure gradient is due to the sudden expansion at the edge of the step-induced flow separation [40]. A sketch of the BFSF 2 in laminar flow is shown in Figure 4.

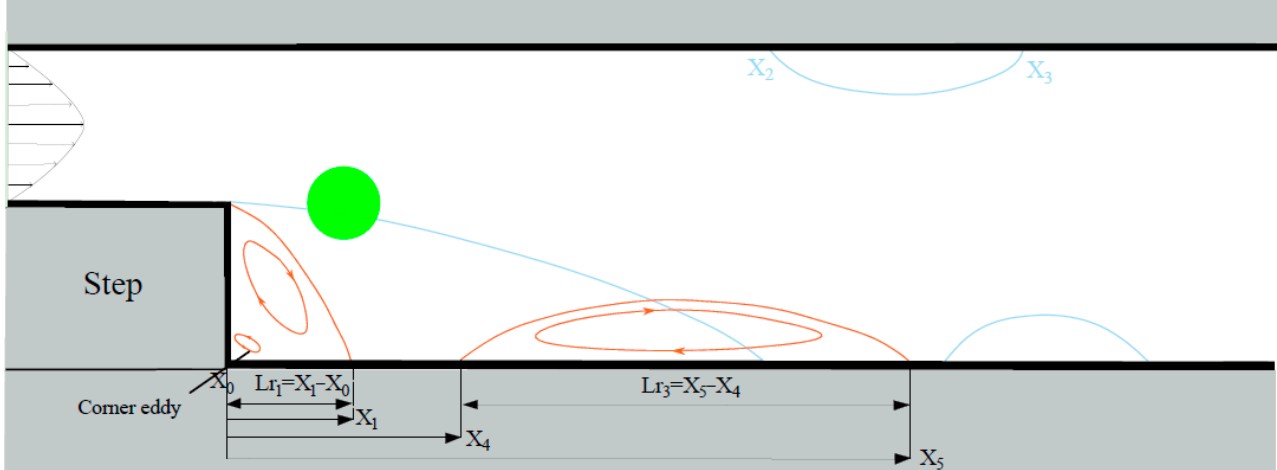

**Figure 4.** Sketch of the flow in the BFSF 2 in laminar flow (Blue lines show the recirculation zones for BFSF 1).

For the BFSF 2, the flow separated at the step, but the dividing streamline was deviated by the cylinder to the bottom wall, and the reattachment point $X_1$ was found to be more upstream than for the BFSF 1 (Figure 4). In addition, the second recirculation zone on the upper wall was missing, while the third recirculation zone was observed even at $75 < Re_h \leq 755$ (Table 4), and it was more upstream than in the BFSF 1. For the BFSF 2, $Lr_1$ and $Lr_3$ increased as $Re_h$ increased. It is important to point out that for the BFSF 2, none of the recirculation zones were observed at $Re_h = 75$. Table 4 lists the value of the normalized location of starting and ending recirculation zones in the BFSF 1 and BFSF 2. For the BFSF 2, while $X_4$ was unchanged, $X_5$, as the $X_5/h$ increased as $Re_h$ increased. The cylinder pushed the primary recirculation region upstream to the corner of the step and, hence, at each $Re_h$, $Lr_1$ was generally lower than that of the BFSF 1.

Figure 5 compares the normalized location of $X_5$ point between the BFSF 2 of the present study and the numerical results of the BFSF 1 of Erturk [41] in various Reynolds numbers.

#### 3.1.2. Cylinder Wake

The flow past a cylinder creates a large drag due to the periodic separation and causes some differences in the pressure between downstream and upstream. Attention was particularly focused on the effect of the cylinder on the 2D flow structure over the backward-facing step. In laminar flow, the structure of the flow past a single cylinder depends on

the cylinder-diameter Reynolds number (Re$_D$) [29]. Figure 6 shows the differences in two-dimensional flow patterns observed as the Reynolds number increased.

**Table 4.** The reattachment and separation points of the recirculation zones vs. Re$_h$ in laminar flow.

| Re$_h$ | $X_1/h$ BFSF1 | $X_1/h$ BFSF2 | $X_2/h$ BFSF1 | $X_2/h$ BFSF2 | $X_3/h$ BFSF1 | $X_3/h$ BFSF2 | $X_4/h$ BFSF1 | $X_4/h$ BFSF2 | $X_5/h$ BFSF1 | $X_5/h$ BFSF2 |
|---|---|---|---|---|---|---|---|---|---|---|
| **75** | 2.88 | - | - | - | - | - | - | - | - | - |
| **158** | 5.25 | 0.45 | - | - | - | - | - | 4.4 | - | 5.26 |
| **336** | 9.15 | 0.7 | 7.8 | - | 10.65 | - | - | 2.3 | - | 12.15 |
| **420** | 10.4 | 0.8 | 8.65 | - | 14.15 | - | - | 2.25 | - | 15.1 |
| **544** | 11.81 | 0.85 | 8.9 | - | 18.6 | - | - | 2.1 | - | 19.5 |
| **672** | 12.65 | 0.9 | 10.2 | - | 21.5 | - | - | 2.1 | - | 23.65 |
| **755** | 13.37 | 0.9 | 10.62 | - | 23.1 | - | - | 2.05 | - | 26.3 |

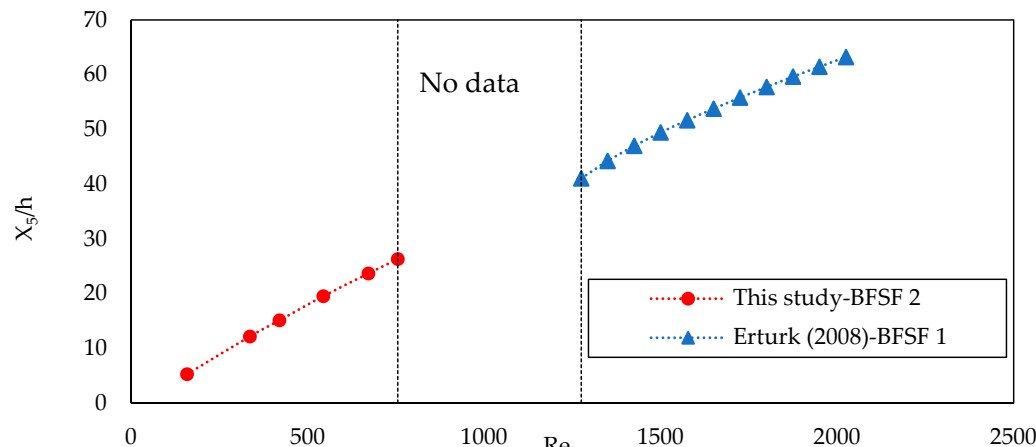

**Figure 5.** Dimensionless the reattachment points of the third recirculation region ($X_5/h$) vs. Re$_h$ [41].

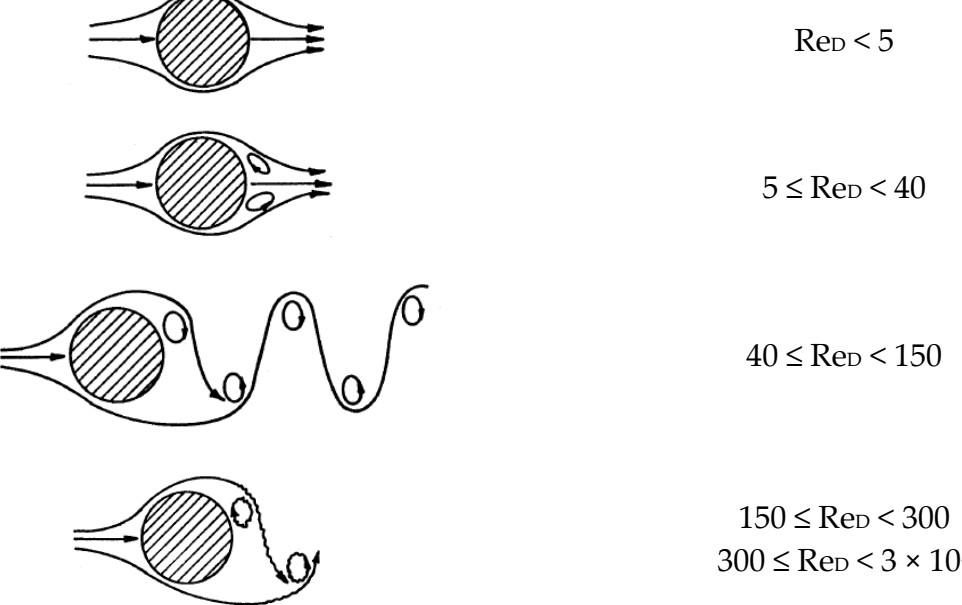

**Figure 6.** Various flow regimes over a 2D cylinder (J.H. Lienhard [29]).

At a low Reynolds number ($Re_D < 5$), the flow remains attached to the cylinder, no separation occurs, and viscous forces are dominant, thereby no wake is formed. For the range $5 \leq Re_D < 40$, a change takes place in the flow patterns and the flow separates from both sides of the cylinder. Two symmetric and stable vortices at both sides are formed and remain attached to the body. As the Reynolds number increased to about $40 \leq Re_c < 150$, the flow pattern is developed. The wake becomes unstable, and one of the two vortices breaks away and then the second is shed alternately from the cylindrical body. This phenomenon, known as Karman Vortex Street, happens because of the flow oscillation and the nonsymmetrical pressure in the wake zone. As the Reynolds number is increased in the range $150 \leq Re_D < 300$, periodic irregular disturbances start in the wake with a gradual transition to turbulent in the vortex wake. Figure 7 depicts a variety of flow patterns downstream of a cylinder in the backward-facing step as the Reynolds number of the fluid increased.

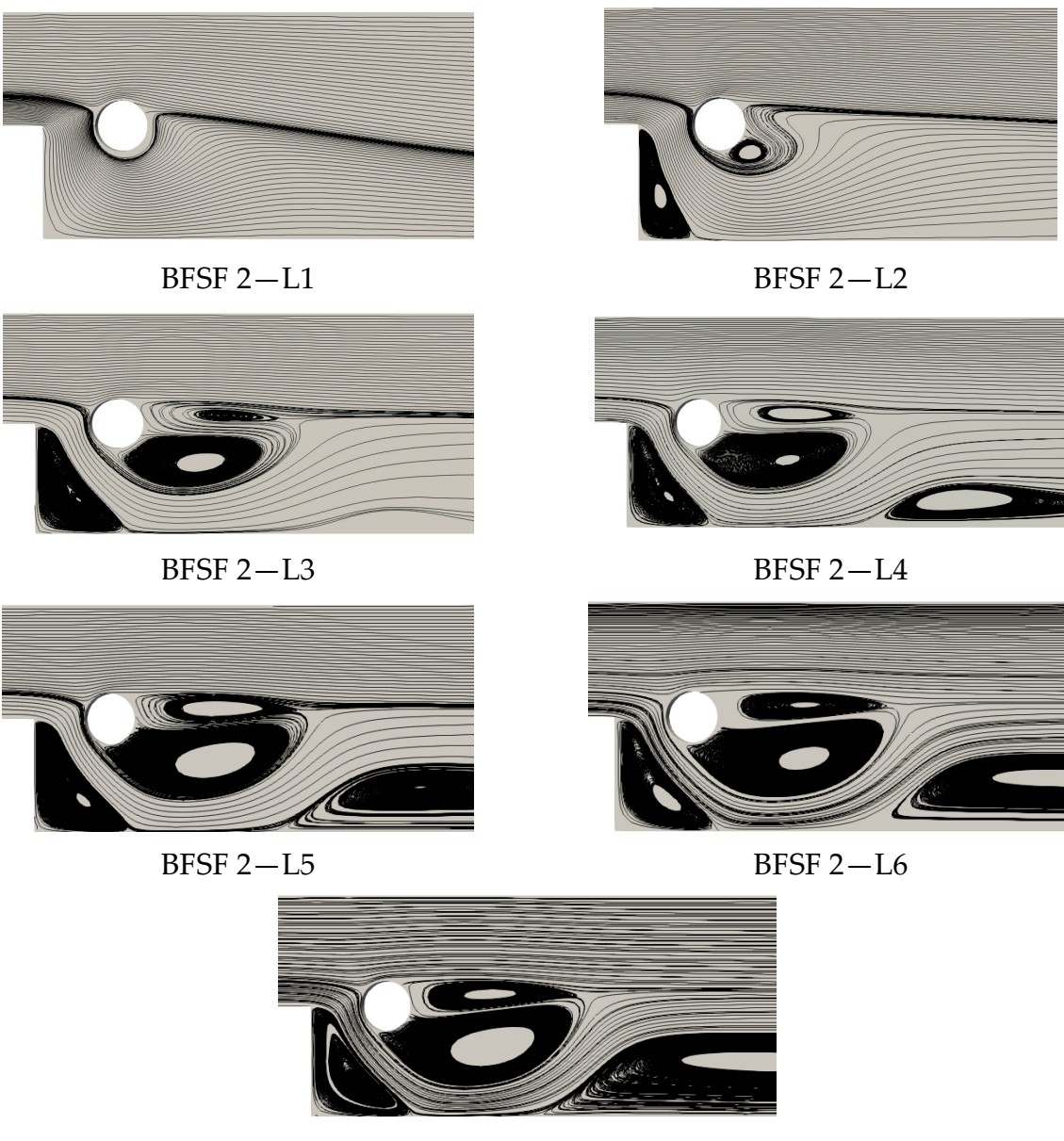

BFSF 2—L1

BFSF 2—L2

BFSF 2—L3

BFSF 2—L4

BFSF 2—L5

BFSF 2—L6

BFSF 2—L7

**Figure 7.** Streamlines of the flow behind a cylinder in the BFSF 2 at different cylinder diameter Reynolds numbers ($Re_D$).

At a low Reynolds number Re$_D$ = 15 (BFSF 2—L1), the flow was not noticeably affected by the presence of the cylinder and vortices did not form behind the cylinder. In the BFSF 2—L2 (Re$_D$ = 28), the streamlines showed one vortex. However, for the range of Reynolds number 5 ≤ Re$_D$ < 40, two symmetric and stable vortices behind a single cylinder were found. For the Reynolds number range Re$_D$ > 40 (BFSF 2—L3, BFSF 2—L4, BFSF 2—L5, BFSF 2—L6, and BFSF 2—L7), two asymmetric vortices are found behind the cylinder in different sizes, and a large portion of these vortices shifted toward the below cylinder. However, for flow past a single cylinder in the Reynolds number range (40 ≤ Re$_D$ < 150), periodic irregular disturbances in the wake of a cylinder were observed. As a cylinder was placed downstream of the step, the step affected the near wake of the cylinder by changing the dynamics of the vortex generation.

### 3.1.3. Vertical Profiles of the Streamwise Velocity

The dimensionless u-velocity (u/U$_{max}$, where U$_{max}$ is the maximum inlet velocity) profiles of the BFSF 1 and BFSF 2 Runs, are shown in Figure 8.

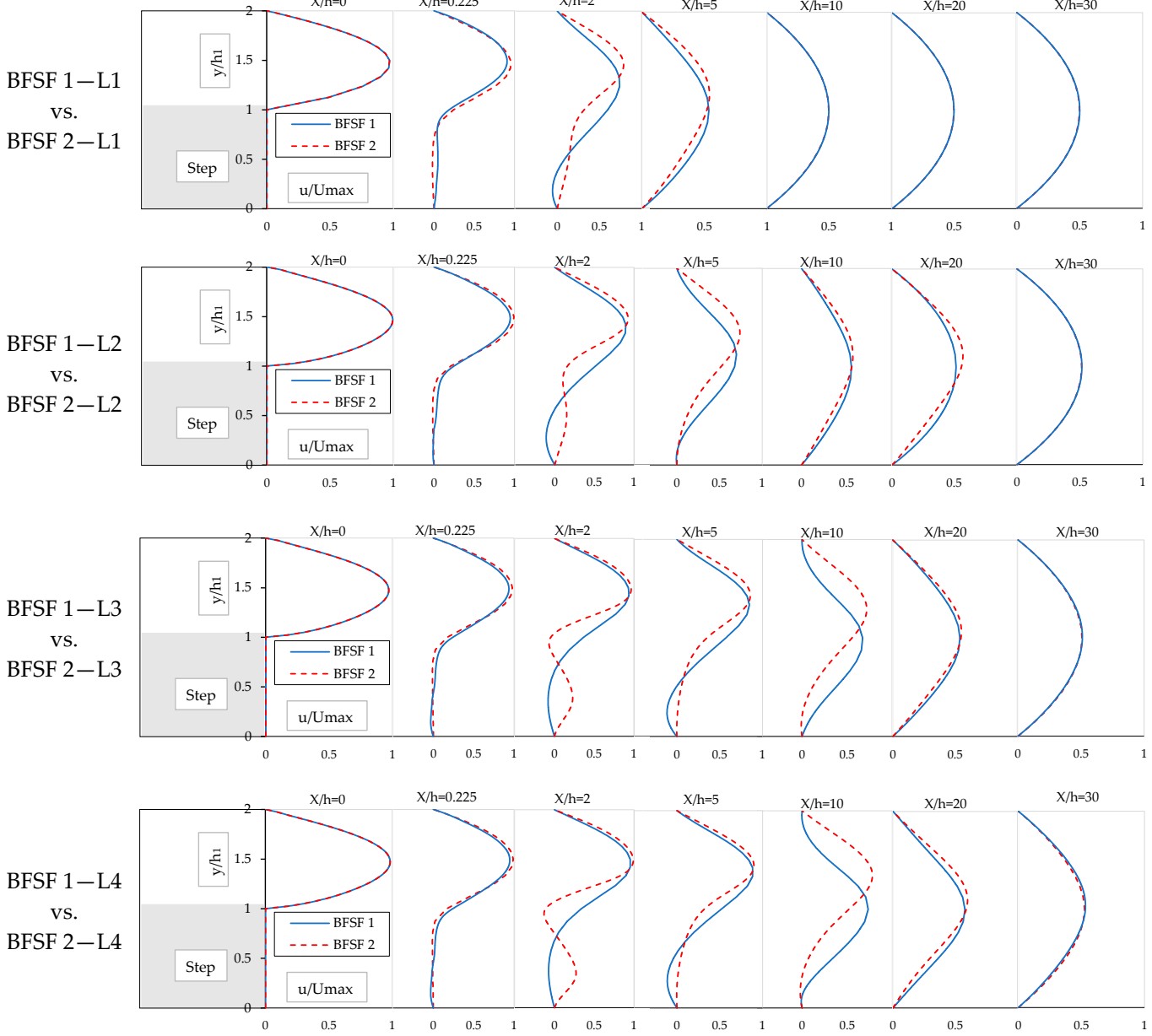

**Figure 8.** *Cont.*

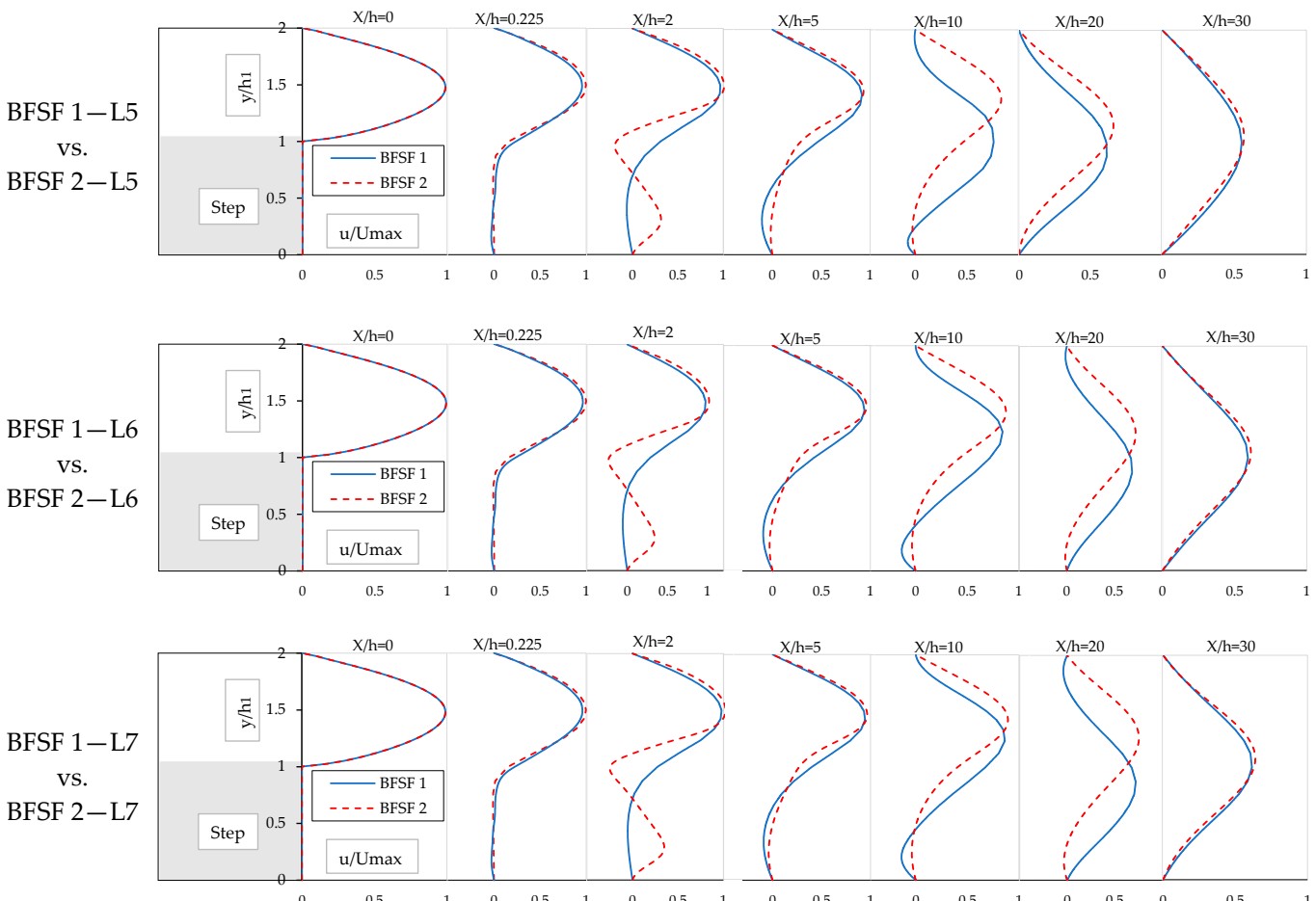

**Figure 8.** Dimensionless u-velocity profiles ($u/U_{max}$) at different Reynolds numbers.

In the BFSF 2, with the incident flow toward the cylinder, the regular patterns of the vortex shed rear of the cylinder. The maximum velocity for the BFSF 2 Runs was a bit higher than that of the BFSF 1, and the location of the maximum velocities shifted toward the upper wall. More importantly, the cylinder increased the skewness of the velocity profiles. The skewness of velocity profiles was calculated for both the BFSF 1 and the BFSF 2 in the following locations: x/h = 2, where the primary recirculation occurred and the velocity distribution was high; x/h = 5, x/h = 10 was downstream of the cylinder; and at x/h = 30 where the flow developed and reached the outlet of the geometry. For the BFSF 2, the percentages of increasing skewness were 15, 185, 110, and 10% at x/h = 2, x/h = 5, x/h = 10, and x/h = 30, respectively. The results indicated that the skewness of the velocity profile was larger near the cylinder than in other locations.

### 3.1.4. Skin Friction Distribution

The distribution of the skin friction coefficient ($C_f$) at the bottom wall was calculated. As shown in Figure 9, the skin friction coefficients of the BFSF 1 and BFSF 2 for different step-height Reynolds numbers were compared. In Figure 9, the vertical dotted line shows the position of the cylinder center.

In the BFSF 1 Runs, the $C_f$ decreased and reached the minimum peaks in the recirculation zone and gradually recovered to positive values downstream of the reattachment point. The constant value skin friction coefficient downstream showed a fully developed channel flow. In the BFSF 2, two peaks of $C_{f,\,min}$ were observed. As previously pointed out, two recirculation zones ($Lr_1$ and $Lr_3$) were observed at the bottom wall of the BFSF 2 in laminar flow. The minimum peak of the skin friction coefficient ($C_{f,\,min}$) occurred due to the recirculating flow where the velocity distribution changed [26]. For the BFSF 2—L1,

the minimum peak of the skin friction coefficient ($C_{f, min}$) was not observed for $Re_h = 75$, revealing the influence of the cylinder on flow features, and hence demonstrating that the recirculation zone was not formed at this Reynolds number. In the other BFSF 2 Runs, the value of $(C_{f, min})_1$ increased, while its position was found to be more upstream than for the BFSF 1. The second minimum peak $(C_{f, min})_2$ was observed far away from the primary one at the bottom wall and its value was smaller than that of the primary $(C_{f, min})_1$.

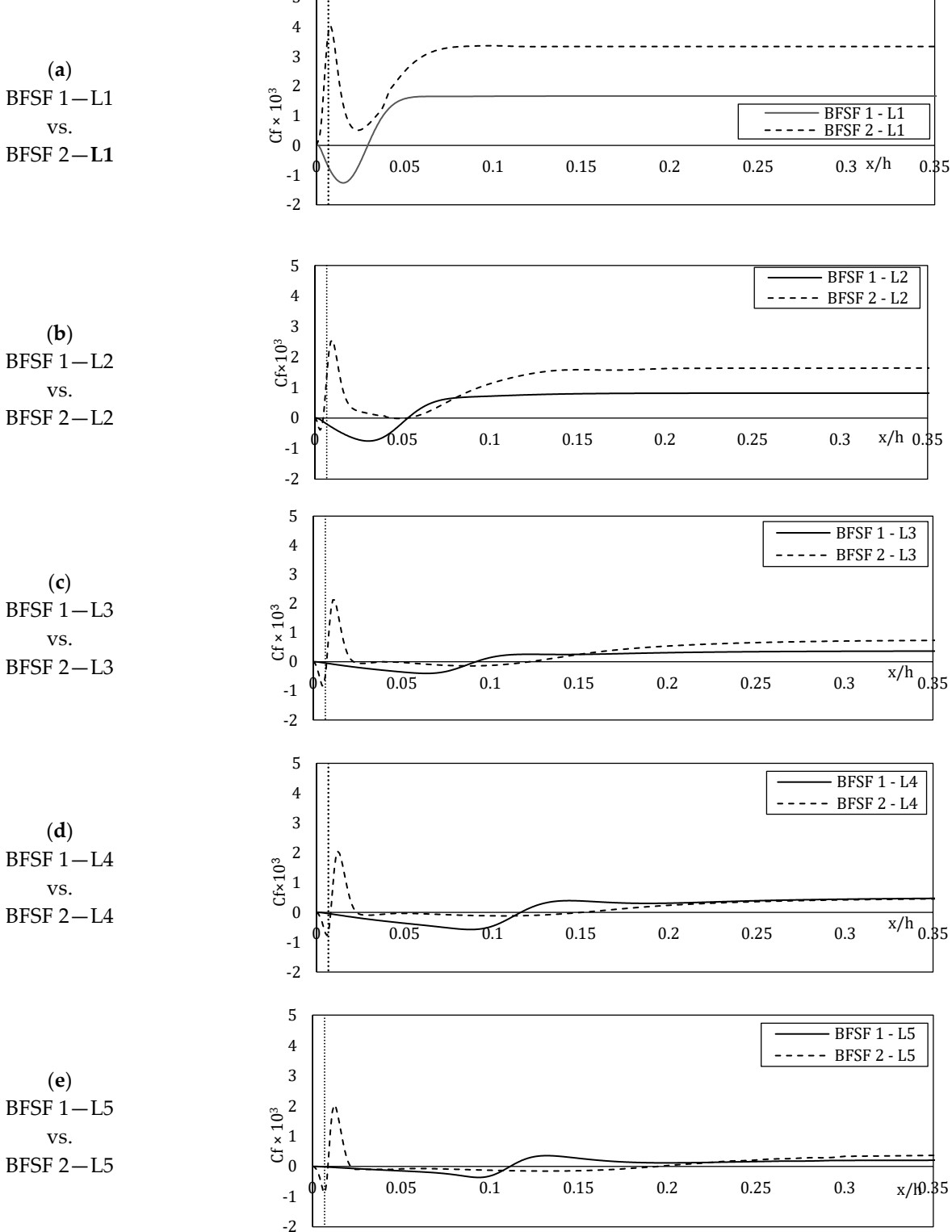

**Figure 9.** *Cont.*

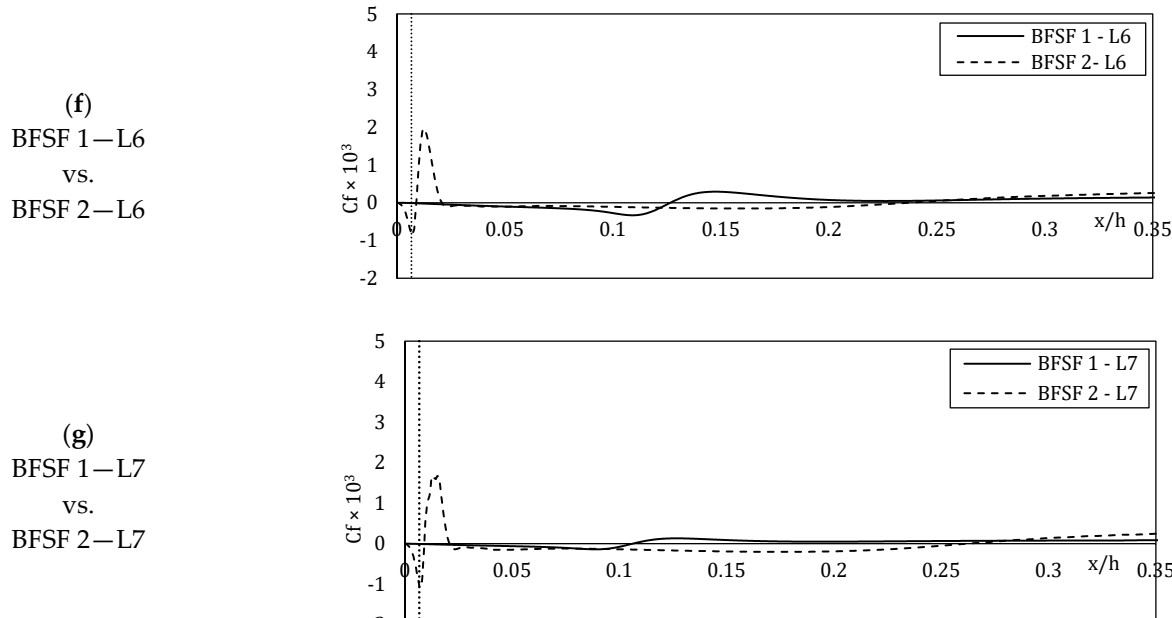

**Figure 9.** Longitudinal distribution of $C_f$ at the bottom wall downstream of the step in the BFSF 1 and BFSF 2 at different $Re_h$.

### 3.2. Turbulent Flow

#### 3.2.1. Recirculation Zone and Reattachment Length

In turbulent flow, the cylinder was installed at different locations downstream of the step. Table 5 lists the value of the reattachment length for the BFSF 1 and BFSF 2 at different locations. The size of the primary recirculation zone increased as the distance of the cylinder increased in the x-direction. Further, it was observed that for BFSF 2—T1, a small third recirculation region ($Lr_3$) formed far away from the primary one on the bottom wall. As previously pointed out, Armaly et al. [40] reported that the third recirculation zone was not found in their study for $Re_h > 1725$. However, for the BFSF 2—T1, the third recirculation zone was observed even for $Re_h = 9000$.

**Table 5.** Reattachment length in the turbulent flow.

|  | BFSF 1—T1 | BFSF 2—T1 | BFSF 2—T2 | BFSF 2—T3 | BFSF 2—T4 | BFSF 2—T5 |
|---|---|---|---|---|---|---|
| **$Lr_1$/h** | 6.75 | 1.1 | 1.54 | 1.90 | 1.46 | 7.81 |
| **$Lr_3$/h** | - | 1.56 | - | - | - | - |

The cylinder pushed the primary recirculation zone upstream to the corner of the step and its length decreased. As previously pointed out, the third recirculation zone was caused by vortex shedding from the edge of the step. In BFSF 2—T1 the flow directly incident cylinder, these vortices were thought to approach the wall, and the third recirculation zone was formed due to the sharp change of flow direction that eddies. In the other Runs, the third recirculation zone was missing.

#### 3.2.2. Cylinder Wake

For the range of cylinder-diameter Reynolds number range $300 < Re_D < 3 \times 10^5$, the flow past a single cylinder developed, and the boundary layers separated from the front stagnation point. There is a fully developed turbulent wake downstream of the cylinder in this range of Reynolds number in flow past a cylinder, the vortex shedding process becomes fully turbulent in the wake, and the vortex street is formed. For $Re_D = 2015$, as the cylinder was placed at different locations downstream of the step, the step affected the near wakes of the cylinder by changing the dynamics of the vortex generation. As shown in Figure 10,

at different locations of cylinders in the horizontal direction (BFSF 2—T1, BFSF 2—T2, and BFSF 2—T3) and in a location of the cylinder above the mid-plane of step (BFSF 2—T4) two recirculation bubbles were observed downstream of the cylinder, with the size of the lower wake recirculation bubble being larger than that of the upper one. As the distance of the cylinder from the edge increased, two vortices behind the circular cylinder were slightly directed downwards. For the BFSF 2—T4 Run, the size of these vortices increased. For the BFSF 2—T5 Run, the flow coming from upstream of the step was not noticeably affected the cylinder and the step did not affect the near wake of the cylinder. Therefore, the separation streamlines from the top corner of the step resembled the counterpart for the unobstructed case.

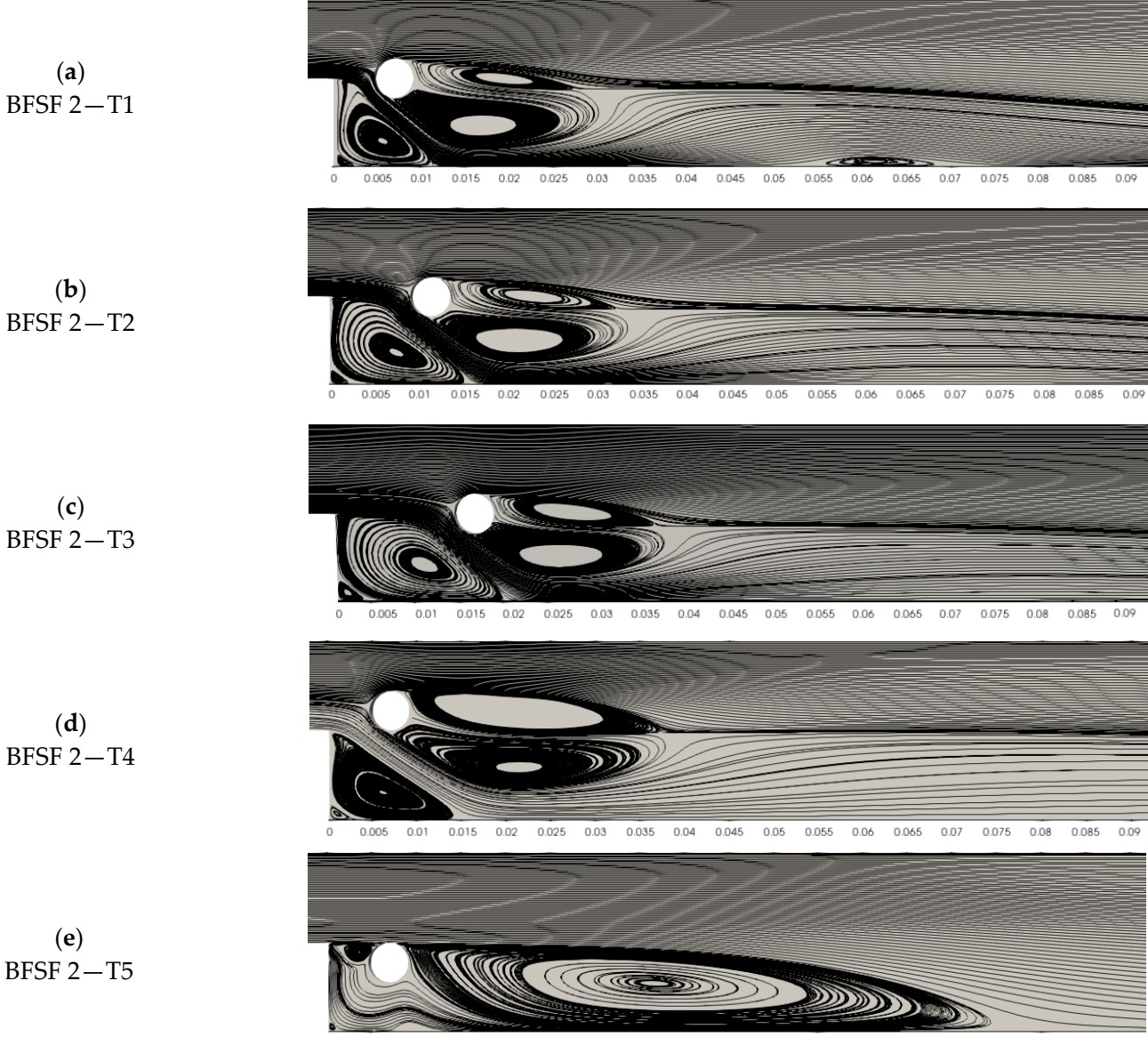

(**a**)
BFSF 2—T1

(**b**)
BFSF 2—T2

(**c**)
BFSF 2—T3

(**d**)
BFSF 2—T4

(**e**)
BFSF 2—T5

**Figure 10.** Streamlines of the flow in BFSF 2 for $Re_D = 2015$ ($Re_h = 9000$).

3.2.3. Vertical Profiles of the Streamwise Velocity

The u-velocity profiles at different locations for the BFSF 1 and BFSF 2 were compared (Figure 11).

For the BFSF 2 Runs, the distribution of vertical profiles of the streamwise velocity was changed. The cylinder affected the regular patterns of the vortex shed to rear of the

cylinder and the location of the maximum velocities shifted toward the upper wall. Further downstream of cylinder and reattachment regions, the flow recovers its fully developed flow behavior. In all Runs, the flow was developed into a backward-facing step toward the outlet.

### 3.2.4. Skin Friction Distribution of the Bottom Wall

The distribution of the skin friction coefficient ($C_f$) at the bottom wall of the BFSF 2 is shown in Figure 12. Note that for the BFSF 2, the location was scaled using $Lr_1$ from the BFSF 1—T1.

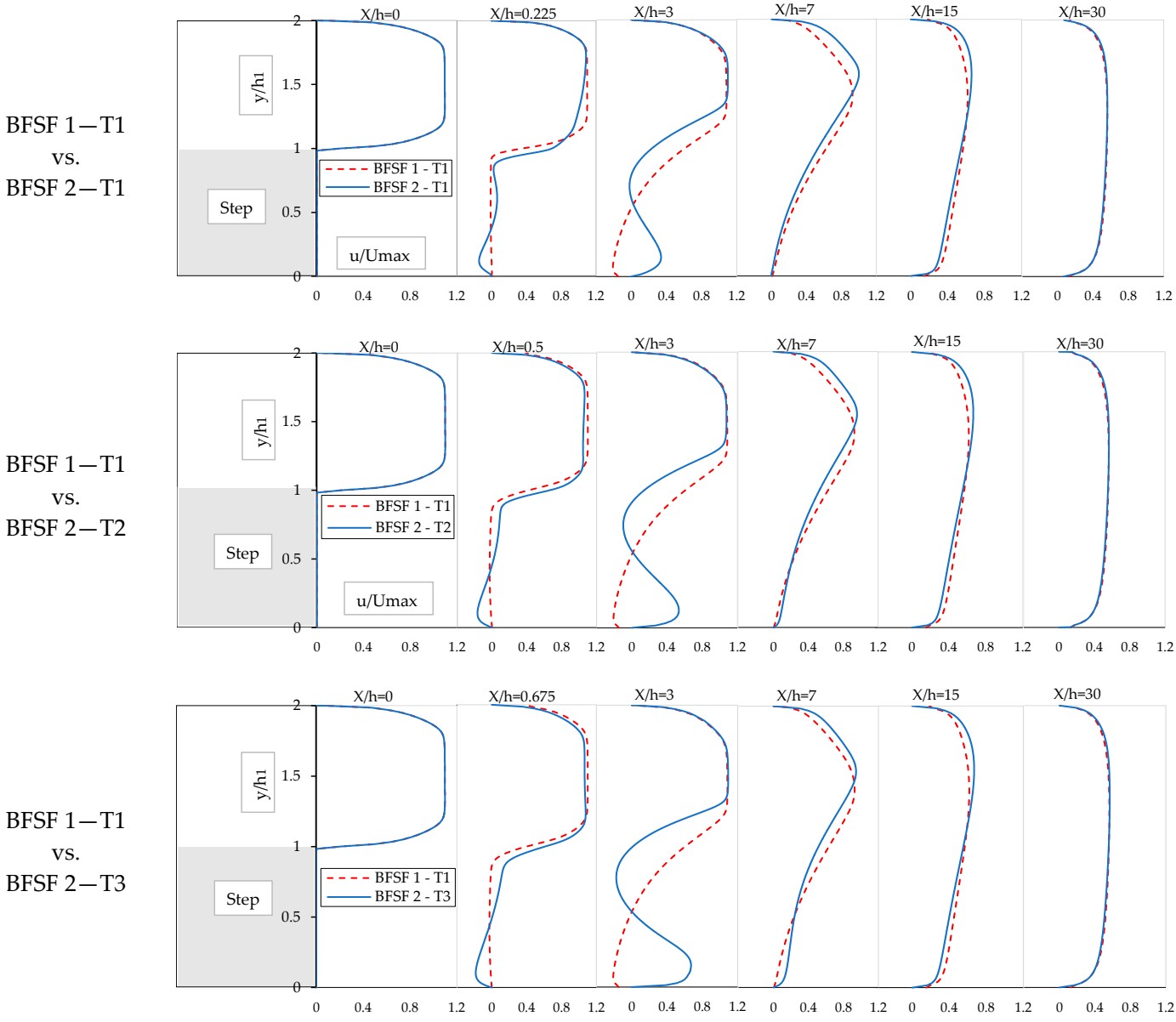

**Figure 11.** *Cont.*

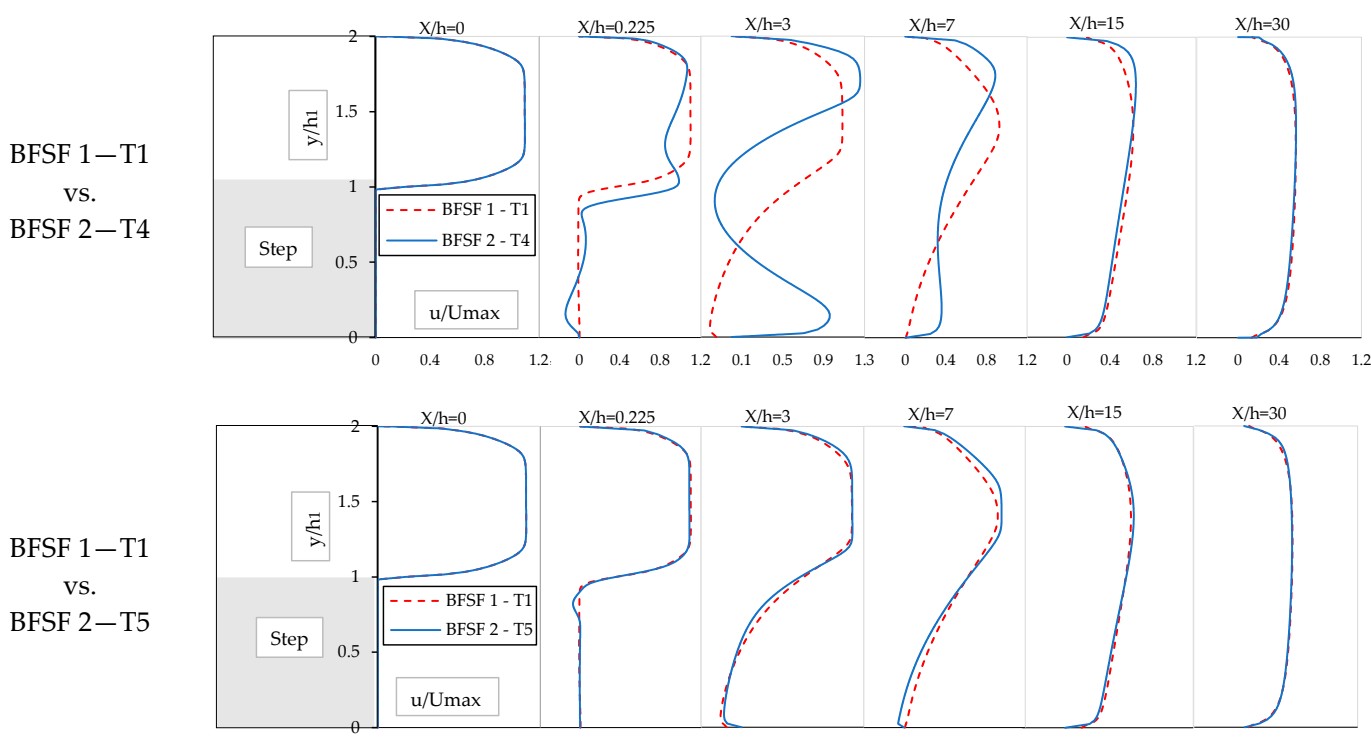

**Figure 11.** Dimensionless u-velocity (u/U$_{max}$) profiles of the BFSF 1 and BFSF 2 at various locations.

For the BFSF 2 Runs, a minimum peak of the skin friction coefficient was observed within the recirculating region [42]. The $C_f$ decreased and reached the minimum peak in the recirculation zone and gradually recovers to positive values downstream of the reattachment point. For the BFSF 2—T5, its behavior was the same as BFSF 1—T1. In the BFSF 2—T5, the minimum values of the skin friction coefficient were lower than those of the BFSF 1—T1. However, for BFSF 2—T1, BFSF 2—T2, BFSF 2—T3, and BFSF 2—T4, two minimum peaks of the skin friction coefficient $(C_{f, min})$ occurred. The value of $(C_{f, min})_1$ increased while its position was found to be upstream than for the BFSF 1—T1. The second minimum peak $(C_{f, min})_2$ was observed far away from the primary one at the bottom wall and its value was smaller than that of the primary $(C_{f, min})_1$. Table 6 lists the values of the minimum peak of $(C_{f, min})_1$, and its location for the BFSF 1 and BFSF 2.

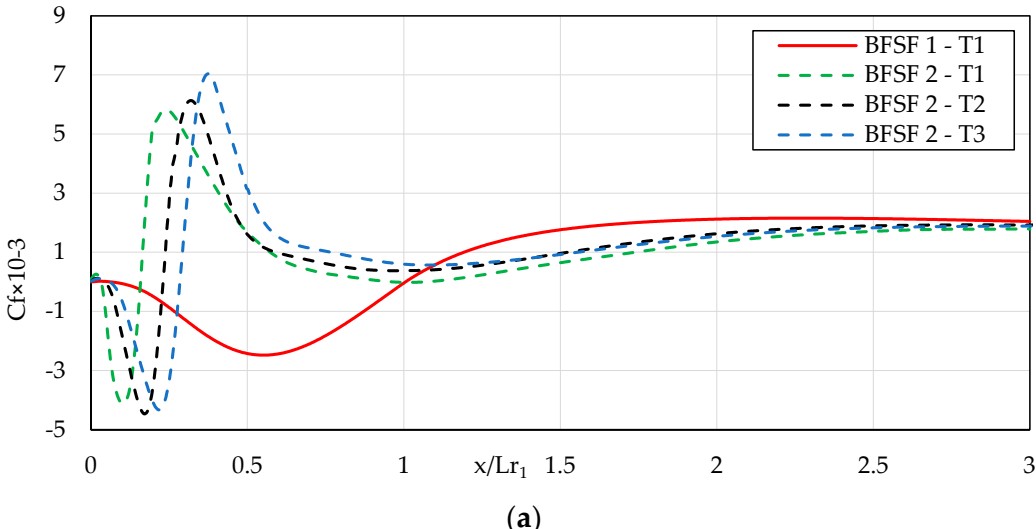

(**a**)

**Figure 12.** *Cont.*

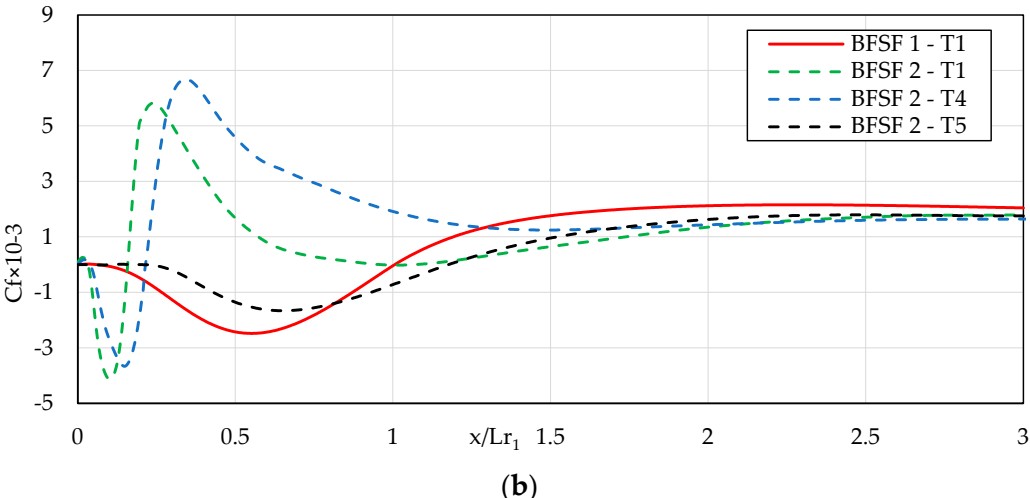

**(b)**

**Figure 12.** Longitudinal distribution of $C_f$ at the bottom wall downstream of the step compared with BFSF 1—T1, (**a**) different locations of the cylinder in the x-direction, and (**b**) different locations of the cylinder in the y-direction.

**Table 6.** The minimum value of skin friction coefficient $(C_{f, min})_1$ and its position $X_{(Cf, min)1}$ in the BFSF 1 and BFSF 2 of turbulent flow.

|  | BFSF1—T1 | BFSF2—T1 | BFSF2—T2 | BFSF2—T3 | BFSF2—T4 | BFSF2—T5 |
|---|---|---|---|---|---|---|
| $-(C_{f,min})_1 \times 10^{-3}$ | 2.48 | 4.12 | 4.33 | 4.47 | 3.66 | 1.66 |
| $X_{(Cf,min)1}/L_{r1}$ | 0.561 | 0.102 | 0.107 | 0.219 | 0.150 | 0.651 |

### 3.2.5. Static Pressure Coefficient of the Bottom Wall

As already performed in Part 1 [2], the normalized pressure coefficient $(C^*_p)$ was used for the comparison of pressure distribution. The normalized pressure coefficients $(C^*_p)$ against the location scaled with the reattachment position, are compared in Figure 13.

In the BFSF 1—T1, the static pressure increased starting from the corner of the bottom wall and a sharp increase of pressure occurred in the reattachment zone (from x = 3 h to x = 7 h). In the BFSF 2 Runs, a sharp increase in pressure occurred in front of the cylinder; however, the pressure behind the cylinder decreased. The distribution of pressure farther downstream remained relatively stable in the flow recovery process.

### 3.2.6. Surface Pressure Distributions of Cylinder

Surface pressure distributions of the cylinder in crossflow, where it was mounted downstream of the step at different locations, are shown in Figure 14.

The step affected the pressure distribution around the cylinder by changing the maximum and minimum points of surface pressure, which moved away from the centerline. The largest pressure was induced on the front side of the cylinder where the incoming flow decelerated while being deflected around the top of the cylinder. The lowest pressures were recorded not at the sides of the cylinder, but rather just at the separation points. As expected, the largest pressure was found for BFSF 2—T4 when the cylinder was positioned above the step and the incoming flow crossed with the cylinder.

### 3.2.7. Turbulent Kinetic Energy

In the RANS turbulence model, the turbulent kinetic energy (k) is given directly by the resolution of its transport equation. Figure 15 shows the distribution of turbulent kinetic energy in the BFSF 1 and BFSF 2 for different locations of the cylinder.

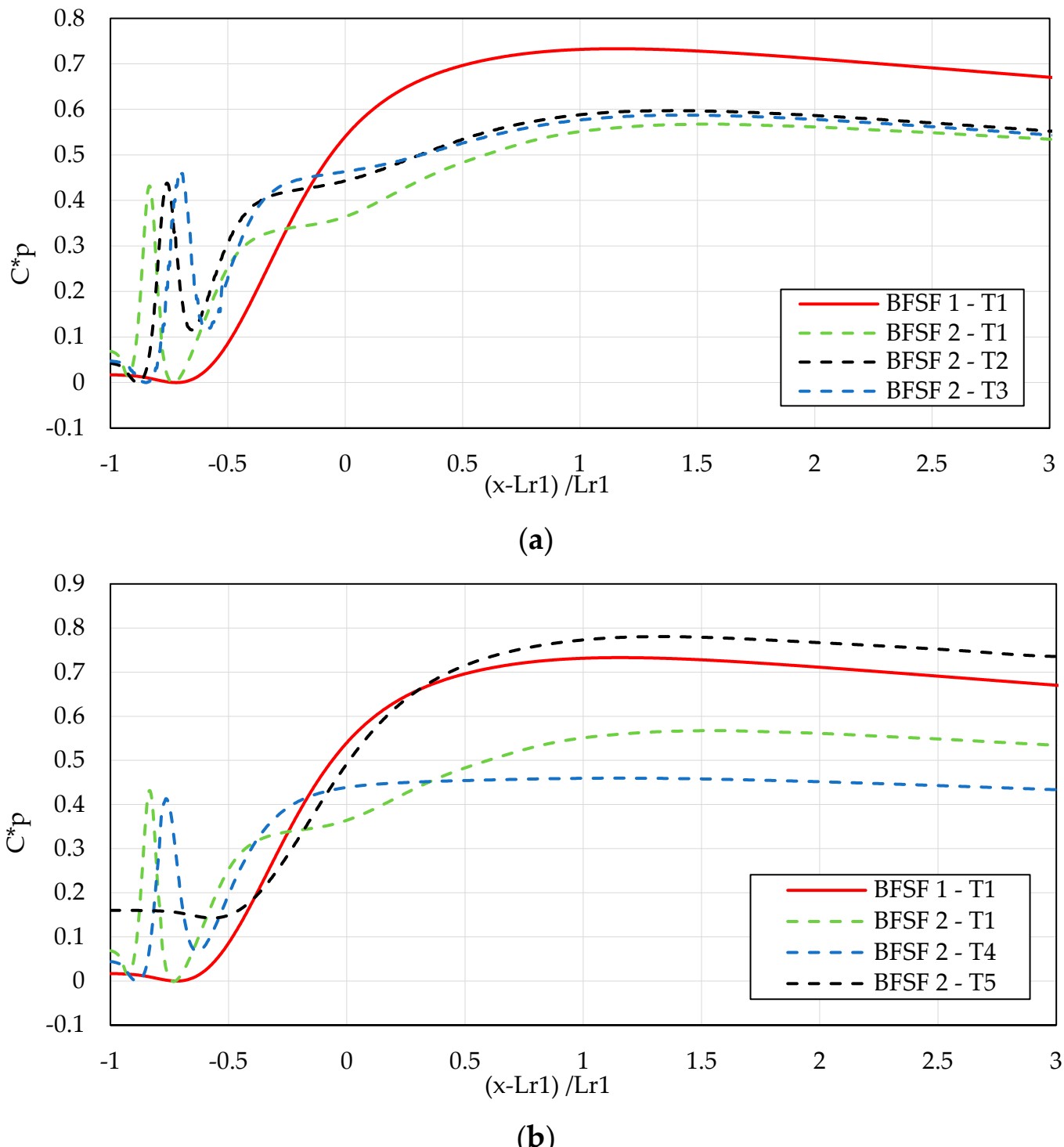

**Figure 13.** Longitudinal normalized pressure coefficient ($C^*_p$) of the bottom wall downstream of the step compare with BFSF 1, (**a**) different locations of the cylinder in the x-direction, and (**b**) different locations of the cylinder in the y-direction.

For BFSF 1—T1 and BFSF 2—T5, the maximum turbulent kinetic energy was below the mid-plane of the step in regions of high shear flow. While, for other the BFSF 2 Runs, the cylinder changed the distribution of turbulent kinetic energy, and the maximum turbulent kinetic energy was shifted above the mid-plane of the step. In the BFSF 1—T1 and BFSF 2—T5, the turbulent kinetic energy decreased monotonically starting from the step edge in the x-direction. However, TKE was amplified downstream of the cylinder in midplane,

and the region of high TKE was also bounded by the cylinder. In the vertical plane, the region of high TKE downstream of the cylinder contained two subregions of high TKE and it was in the highest value when the cylinder was above the mid-plane of step (BFSF 2—T4). These subregions were even better delimited in Figure 16 showing the value of maximum turbulent kinetic energy profiles. It is noted that the profiles of $(k_{max}/(U_{max})^2$ were measured in a vertical plane in the section where the maximum value of turbulent kinetic energy was found. The top subregion of high flow turbulence was mostly due to the passage of flows inside the separated shear layers of the cylinder.

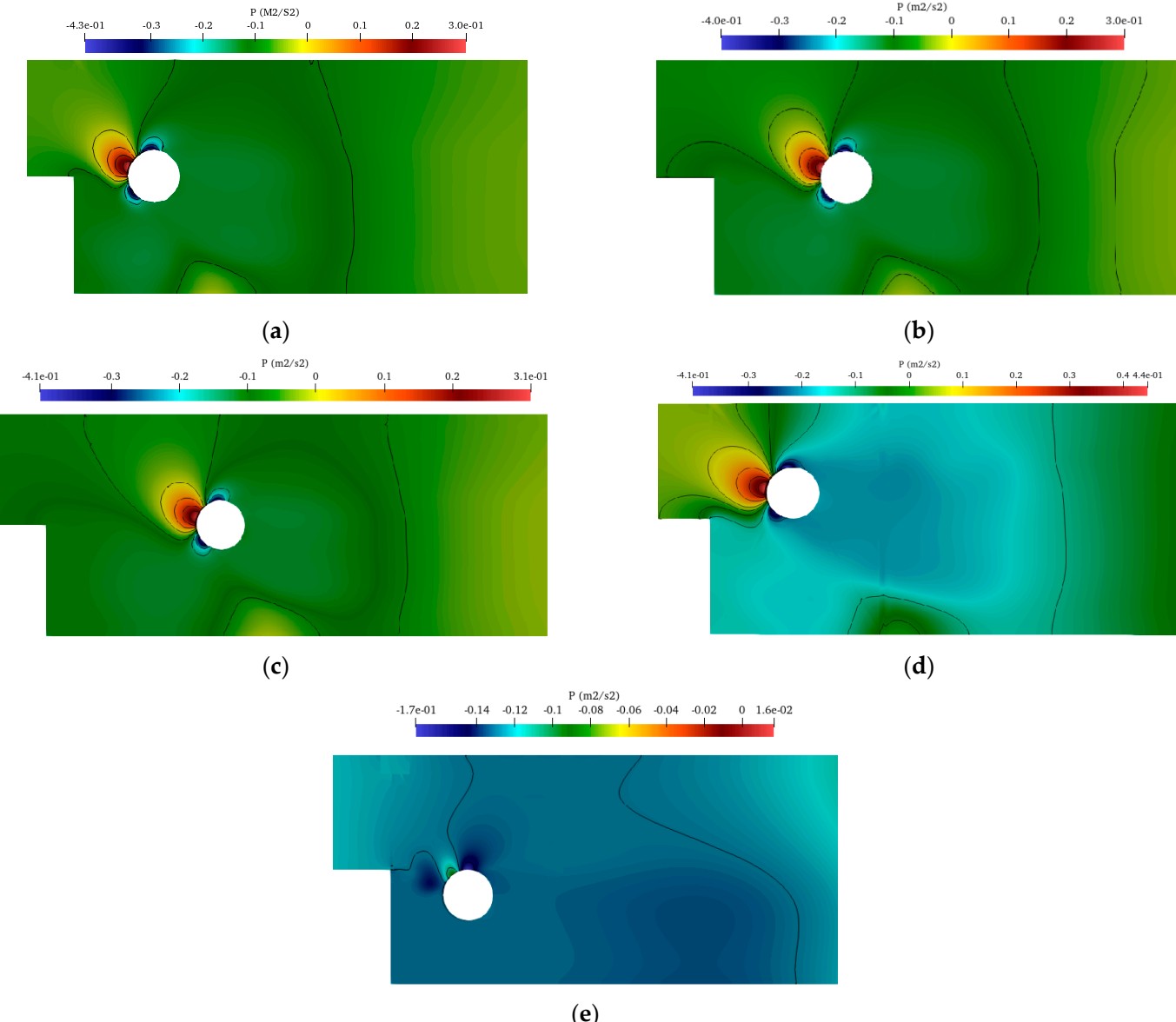

**Figure 14.** Distribution of surface pressure of cylinder for (**a**) BFSF 2—T1, (**b**) BFSF 2—T2, (**c**) BFSF 2—T3, (**d**) BFSF 2—T4, and (**e**) BFSF 2—T5.

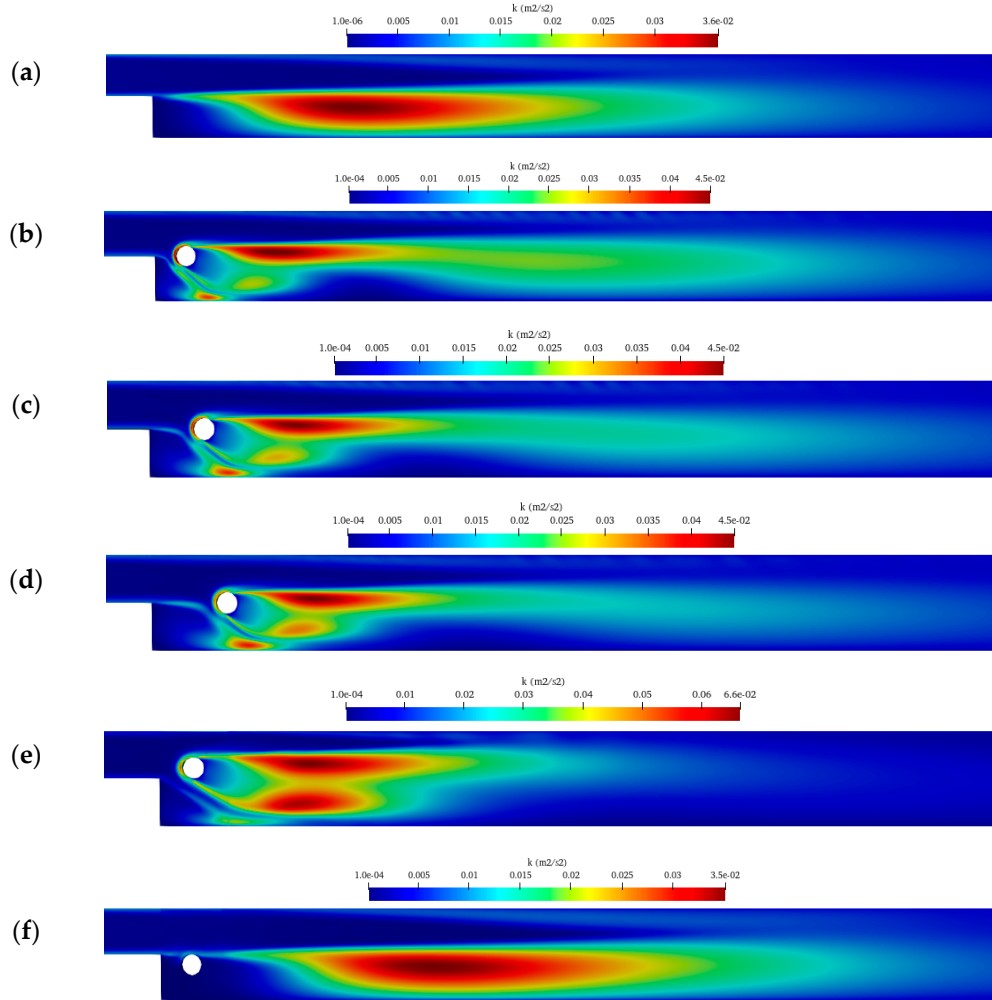

**Figure 15.** Distribution of turbulent kinetic energy (k) for (**a**) BFSF 1—T1 (**b**) BFSF 2—T1 (**c**) BFSF 2—T2 (**d**) BFSF 2—T3 (**e**) BFSF 2—T4 (**f**) BFSF 2—T5.

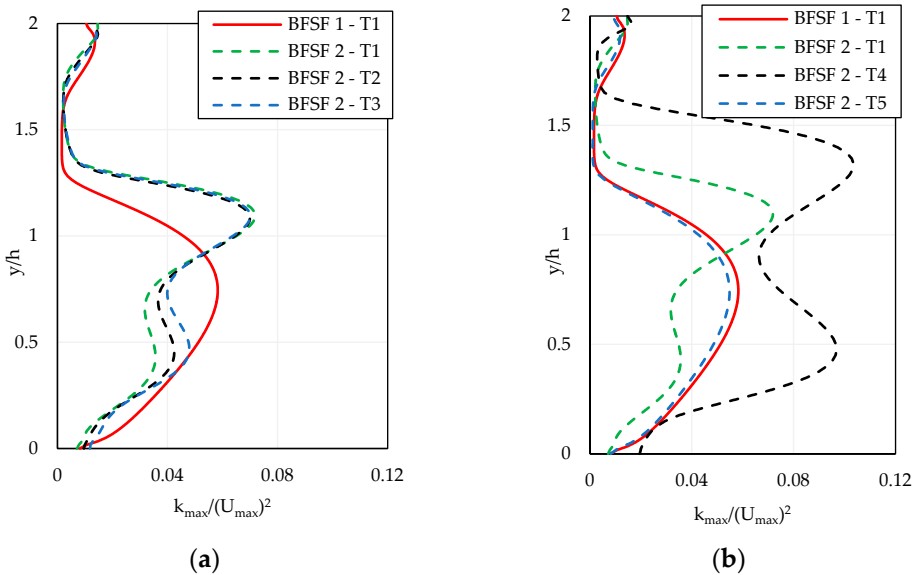

**Figure 16.** Dimensionless maximum turbulence kinetic energy downstream of the step compared with BFSF 1, (**a**) different locations of the cylinder in the x-direction, and (**b**) different locations of the cylinder in the y-direction.

## 4. Discussion

Flow downstream of the step is complex and the presence of a cylinder creates eddies, transverse flows, velocity gradients, and other spatial flow patterns. A better understanding of how the cylinder interacts to create spatially varying flows downstream of the step leads to the quantification of its features. The key findings from this study are as follows:

**Recirculation zone:** In the BFSF 1, three recirculation zones were observed: the first recirculation zone on the lower wall in laminar and turbulent regimes; the second recirculation zone at the upper wall for $Re_h > 300$; and the third recirculation zone on the lower wall in the early part of the transitional regime. In laminar flow, the cylinder pushed the primary recirculation region upstream to the corner of the step and its length decreased, while the second recirculation zone near the upper wall was missing for BFSF 2. In the BFSF 2, the third recirculation zone was observed even for laminar and turbulent flow when a cylinder was positioned at a diameter distance from the step edge and its location was more upstream than in the BFSF 1. In turbulent flow, the size of the third recirculation zone was smaller than that of laminar flow. As the cylinder was placed far away from the step and above or below the step, the third recirculation zone was missing.

**Cylinder wake:** In laminar flow, the step modified the 2D flow structure past the cylinder, leading to an asymmetric wake distribution. A large portion of these vortices shifted toward the below cylinder. In turbulent flow, when a cylinder was positioned along the step edge or above the step edge, flow passing over the cylinder suppressed the formation of the von Kármán vortex street, two vortices formed behind the cylinder in different sizes, and their location shifted towards the bottom wall. As the cylinder was located below the step, its behavior was the same as in BFSF 1.

**Streamwise velocity:** The cylinder increased the velocity due to a narrow cross-section downstream of the cylinder. The location of the maximum velocity shifted towards the middle of the channel in both laminar and turbulent regimes.

**Skin friction distribution:** The wall shear stress is associated with the skin friction coefficient at the bottom wall. A minimum value of skin friction coefficient ($C_{f, min}$) at the bottom wall occurred due to the recirculating flow. In the BFSF 1, a minimum value of skin friction coefficient ($C_{f, min}$) at the bottom was observed in both laminar and turbulent flow. However, in the BFSF 2, two minimum peaks of skin friction coefficient ($C_{f, min})_1$ and ($C_{f, min})_2$ were observed due to the two recirculation zones for $Re_h > 75$. The cylinder downstream of the step produced significantly higher minimum and maximum values of the skin friction coefficient at the bottom wall than that without the cylinder.

**Pressure distribution:** The cylinder affected the distribution of pressure along the bottom wall. In the BFSF 2, the minimum and maximum values of the pressure coefficients were lower than those in the BFSF 1. However, the average value of pressure coefficients downstream of the reattachment point was smaller than that in the BFSF 2. In addition, the step affected the distribution of the surface pressure of the cylinder by moving the largest pressure region to the top of the cylinder.

**Turbulent kinetic energy:** In the BFSF 1, the maximum turbulent kinetic energy was found downstream of the step, below the mid-plane of the step. However, in the BFSF 2, the cylinder increased the turbulent kinetic energy and the location of the maximum TKE shifted toward the centerline of the channel. The highest regions of TKE were found in the wakes of the cylinder and its value was higher than that of BFSF 1.

## 5. Conclusions

In the present study, two geometries were comparatively considered, namely the classical BFSF (BFSF 1) and a BFSF with a cylinder placed downstream of the step (BFSF 2), to investigate in both laminar and turbulent flow how the cylinder modifies the 2D classical BFS flow structure using the open-source code OpenFOAM.

When a cylinder was placed downstream of the step, in both laminar and turbulent flow, the added cylinder significantly modified the structure of the recirculating flow over the BFS. This led to an increase in the skin friction coefficient and pressure coefficient in

the recirculation zone. The skin friction coefficient and pressure distribution downstream of the reattachment point remained stable in the flow recovery process. Additionally, in turbulent flow, turbulent kinetic energy increased downstream of the cylinder in the wake of the cylinder. Furthermore, the step influenced the dynamics of the vortex generation and shedding which, in consequence, led to an asymmetric wake distribution in the laminar and turbulent flow.

As in this study, the geometry was a backward-facing step flow, follow-on studies are needed to analyze the effect of cylindric obstacles on flow and turbulence characteristics in a step open-channel. In addition, more studies are also needed to analyze the effects of a group of cylinders over the step.

**Author Contributions:** This paper was created under the joint effort of all authors. Conceptualization, C.G.; Methodology, M.A. and C.G.; Software, Validation, Formal Analysis and Investigation, M.A.; Data Curation M.A. and C.G.; Writing—Original Draft Preparation, M.A.; Writing—Review and Editing, C.G., P.G. and D.F.V.; Visualization M.A.; Supervision, C.G., P.G. and D.F.V. All authors have read and agreed to the published version of the manuscript.

**Funding:** This research received no external funding.

**Institutional Review Board Statement:** Not applicable.

**Informed Consent Statement:** Not applicable.

**Data Availability Statement:** The datasets generated during and/or analyzed during the current study are available from the corresponding author upon reasonable request.

**Acknowledgments:** We are thankful for the insightful comments offered by Andrea Vacca, University of Naples Federico II. The first author acknowledges the financial support from the PhD Program in "Civil Systems Engineering" of University of Napoli Federico II.

**Conflicts of Interest:** The authors declare no conflict of interest.

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
