# Peer review of "Numerical Study of Flow Downstream a Step with a Cylinder Part 2: Effect of a Cylinder on the Flow over the Step"

_fluids, doi:10.3390/fluids8020060_

Round 1
Reviewer 1 Report
The authors presented a Numerical study of Flow Downstream a Step with a Cylinder.
The first part of the work was dedicated to the validation of the numerical models and the results are presented in Part 2.
The authors must explain why the work is divided to 2 parts. Is it necessary to present the validation in a separate paper ?
For Part 1:
The main findings are to be mentioned in the abstract.
The introduction is relatively short and should be extended.
The governing equations and boundary conditions are to be presented before the mesh generation.
Why the buoyancy forces exist in the laminar formulation and doesn’t exist in turbulent formulation.
How the buoyancy terms are treated ? have you considered the Boussinesq approximation ? to be explained.
How the pressure terms are treated ?
More details on the numerical methods are to be provided.
What is the convergence criterion?
The authors considered time depending governing equations, thus the time at which the results are presented is to be indicated.
A 2D validation/verification (2D flow structure for example) is to be added.
Avoid the green background used in some figures
How can you explain the notable differences in Fig 11?
For Part 2:
some quantitative results are to be added to the abstract.
The introduction is to be extended.
The novelty of the work is to be clearly stated.
The used turbulence model is to be justified
A figure presenting the used mesh is to be added
A grid sensitivity test is to be performed.
Arrows are to be added on the streamlines for a better understanding of the flow’s directions (Figs 6 and 9).
The legend of fig 13, is very small.
Why the results are not limited to the more accurate models, found in part 1?
Author Response
The authors would like to thank the reviewers for the useful comments and believe that they provided the opportunity to improve the quality of the work. Every attempt has been made to address and incorporate the comments accordingly.

Reviewer 2 Report
I do not understand why there is two parts. This work should be in the same pdf than the part 1
Line 61 -- brackets
Mesh with cylinder should be presented
More explanation about why those Re are used is needed (Why 9 000¿?)
In the paper is indicated that the number of cells is kept equal than part 1. How this is possible if part 1 is without cylinder?
line 98 --- . dot
I do not see the utility of figure 6
Figure6 L5, L6 and L7 what is happening in the zone where there is no streamlines?
Figure 9 . Use an axis. It would be better for vortex lenght comparison.
Figure 13 Use dimensionless P and same scale.
Figure 15. Use same scale.
Conclusion. As I see it, I should extend much more and give more details about the results
Author Response

(The authors gave the same response as above.)

Reviewer 3 Report
Please see the attachment.

Author Response

(The authors gave the same response as above.)

Round 2
Reviewer 1 Report
After revision, the paper can be accepted for publication
Reviewer 3 Report
The authors have addressed all my comments/suggestions. I found their responses quite satisfactory and the revised version has been much improved. I now recommend the paper for publication. Well-done.